# Afatinib Treatment Alone or with Bevacizumab in a Real-World Cohort of Non-Small Cell Lung Cancer Patients with Epidermal Growth Factor Receptor Mutation

**DOI:** 10.3390/cancers14020316

**Published:** 2022-01-09

**Authors:** Chih-Hsi Scott Kuo, Tzu-Hsuan Chiu, Pi-Hung Tung, Chi-Hsien Huang, Jia-Shiuan Ju, Allen Chung-Cheng Huang, Chin-Chou Wang, Ho-Wen Ko, Ping-Chih Hsu, Yueh-Fu Fang, Yi-Ke Guo, Cheng-Ta Yang

**Affiliations:** 1Division of Thoracic Oncology, Department of Thoracic Medicine, Chang Gung Memorial Hospital, Chang Gung University, College of Medicine, Taoyuan 333, Taiwan; chihhsikuo@gmail.com (C.-H.S.K.); kodochasana@gmail.com (T.-H.C.); seintseint@gmail.com (P.-H.T.); b9502072@cgmh.org.tw (C.-H.H.); b9502008@cgmh.org.tw (J.-S.J.); mr0818@cgmh.org.tw (A.C.-C.H.); howwenko@gmail.com (H.-W.K.); 8902049@gmail.com (P.-C.H.); fang2002@ms68.hinet.net (Y.-F.F.); yang1946@adm.cgmh.org.tw (C.-T.Y.); 2Thoracic Oncology Unit, Chang Gung Memorial Hospital Cancer Center, Taoyuan 333, Taiwan; 3Data Science Institute, Department of Computing, Imperial College London, London SW7 2AZ, UK; y.guo@imperial.ac.uk; 4Division of Pulmonary & Critical Care Medicine, Kaohsiung Chang Gung Memorial Hospital, Kaohsiung 833, Taiwan

**Keywords:** NSCLC, *EGFR* mutation, afatinib, bevacizumab, real-world

## Abstract

**Simple Summary:**

Previous studies of first-generation EGFR-TKI erlotinib and bevacizumab combination have demonstrated superior treatment efficacy compared to erlotinib monotherapy for advanced *EGFR*-mutant NSCLC patients. Whether this combination benefit can also be observed in second-generation EGFR-TKI afatinib-treated patients remains unclear. The study presented a real-world cohort of advanced NSCLC patients with *EGFR* mutation treated by afatinib plus bevacizumab or single-agent afatinib. After balancing the key characteristics between the two treatment groups, the result showcased a similar therapeutic efficacy of afatinib plus bevacizumab compared to afatinib monotherapy. The incidence of drug-resistant mutation was also similar between the two groups. This study provided a clinical practice-based evidence that the additional benefit of bevacizumab is likely moderate in afatinib-treated patients.

**Abstract:**

Background: Treatment outcome between afatinib alone or with bevacizumab in non-small cell lung cancer (NSCLC) patient with epidermal growth factor receptor (*EGFR*) mutation remains insufficiently reported. Methods: A total of 405 advanced NSCLC patients with sensitizing-*EGFR* mutation receiving first-line single-agent afatinib or with bevacizumab were grouped and propensity score-matched. Progression-free survival (PFS), overall survival (OS) and secondary T790M mutation were analyzed. Results: In the original cohort, 367 (90.6%) patients received afatinib treatment alone and 38 (9.4%) patients received afatinib plus bevacizumab. Patients who received bevacizumab combination were significantly younger (54.6 ± 10.9 vs. 63.9 ± 11.5; *p* < 0.001) compared to the afatinib alone group. After propensity score matching, the afatinib alone and afatinib plus bevacizumab groups contained 118 and 34 patients, respectively. A non-significantly higher objective response was noted in the afatinib plus bevacizumab group (82.4% vs. 67.8%; *p* = 0.133). In the propensity score-matched cohort, a bevacizumab add-on offered no increased PFS (16.1 vs. 15.0 months; *p* = 0.500), risk reduction of progression (HR 0.85 [95% CI, 0.52–1.40]; *p* = 0.528), OS benefit (32.1 vs. 42.0 months; *p* = 0.700), nor risk reduction of death (HR 0.85 [95% CI, 0.42–1.74] *p* = 0.660) compared to the single-agent afatinib. The secondary T790M rate in afatinib plus bevacizumab and afatinib alone groups was similar (56.3% vs. 49.4%, *p* = 0.794). Multivariate analysis demonstrated that *EGFR* L858R (OR 0.51 [95% CI, 0.26–0.97]; *p* = 0.044), *EGFR* uncommon mutation (OR 0.14 [95% CI, 0.02–0.64]; *p* = 0.021), and PFS longer than 12 months (OR 2.71 [95% CI, 1.39–5.41]; *p* = 0.004) were independent predictors of secondary T790M positivity. Conclusion: Bevacizumab treatment showed moderate efficacy in real-world, afatinib-treated NSCLC patients with *EGFR*-sensitizing mutation.

## 1. Introduction

EGFR-TKI administration for advanced *EGFR*-mutated NSCLC serves as the major standard of care in the front-line treatment setting. Resistance to the therapy, however, almost inevitably happens at approximately 9 to 14 months, respectively, with first- and second-generation EGFR-TKI [1,2] and at 19 months with third-generation EGFR-TKI [3]. Many combination strategies with different therapeutic modalities have been attempted to improve the treatment efficacy with variable degree of success [4].

The anti-angiogenesis agent bevacizumab in association with erlotinib has been one of the most investigated combination strategies. The first-generation EGFR-TKI erlotinib, as an EGFR/ErbB1-selective inhibitor with a reversible and non-covalent binding property [5], likely possesses an anti-tumor activity more vulnerable to intra-tumoral concentration variation as a result of the impaired drug delivery through a pathologic vasculature of tumor stroma [6,7]. The treatment of vascular endothelial growth factor (VEGF) blocking agent bevacizumab potentially remodels the pathologic and disorganized tumor vessels [8] and in part accounts for the synergistic effect of the combination. In addition, the extensive cross-talk between EGFR and VEGF signaling pathways which collaboratively facilitate tumorigenesis further justifies the joint targeting strategy [9].

Therefore, previous randomized clinical trials involving Asian advanced NSCLC patients with *EGFR* mutation have demonstrated that bevacizumab plus erlotinib offered a higher efficacy, in terms of PFS, compared to erlotinib treatment alone. However, the PFS benefit did not translate to an OS benefit with the add-on of bevacizumab in these studies [10,11,12]. On the other hand, another clinical trial which mainly involved Caucasian patients revealed that the addition of bevacizumab to erlotinib provided neither PFS nor OS benefit compared to erlotinib treatment alone [13]. Aside from erlotinib, the combination of gefitinib and bevacizumab has also been explored in a small phase II study showing a manageable toxicity profile whereas the efficacy did not meet the pre-specified endpoint [14]. Given the inconsistent clinical trial results and the patient representation to clinical practice settings, data from large cohorts of real-world patients are necessary to supplement the knowledge of this combination approach.

The second-generation EGFR-TKI afatinib, with a higher potency and a broader spectrum of ErbB family suppression, has demonstrated a superior treatment efficacy than the first-generation EGFR-TKIs in both clinical trial and real-world settings, albeit at the expense of a higher toxicity profiles [1,15,16,17]. Many attempts have been made to investigate the combination of afatinib with bevacizumab for the first-line treatment of *EGFR*-mutated NSCLC. A manageable adverse event and a promising efficacy were observed in some early-phase clinical trials [18,19] and a satisfactory tolerability and treatment outcome were also reported in real-world patients [20]. Nevertheless, whether the combination of afatinib and bevacizumab presents a synergistic effect and, thus, outperforms the single-agent afatinib in a front-line setting, remains largely unknown.

In addition, blocking of VEGF pathway by bevacizumab or vandetanib has been demonstrated to be effective against mouse xenograft model harboring an *EGFR* T790M mutation [21,22] and a high effectiveness of erlotinib plus bevacizumab was also observed in *EGFR*-mutated NSCLC patients presenting a de novo T790M [23]. Nevertheless, findings from clinical trial showed that the development of acquired T790M after single-agent erlotinib or with anti-angiogenesis agent did not differ significantly [24,25]. Data regarding whether the acquired T790M rate differs between afatinib plus bevacizumab and afatinib monotherapy remains largely unavailable.

In the present study, a large cohort of real-world advanced NSCLC patients with *EGFR* mutation treated by first-line afatinib alone or with bevacizumab was retrospectively investigated. The patient characteristics, treatment outcome, and development of secondary T790M between the two groups were analyzed.

## 2. Methods

### 2.1. Patients and Treatment

We performed a retrospective analysis between January 2014 and December 2019, during which patients receiving afatinib treatment alone or afatinib plus bevacizumab as the first-line treatment of advanced NSCLC with non-resistant *EGFR* mutation were reviewed. All patients in the present study received a starting dose of 40 mg/day of afatinib and those who received the treatment less than one week were excluded. In patients who received a combination of afatinib and bevacizumab, the dose of bevacizumab was administered at 7.5 mg/kg every 3 weeks. Patients were excluded if the first dose of bevacizumab was given 3 weeks behind the first dose of afatinib treatment. The progression-free survival (PFS) was defined as the interval between the date of starting afatinib and the date of radiologically documented progression or death. The treatment response, including complete response (CR), partial response (PR), stable disease, and progressive disease, was evaluated according to the Response Evaluation Criteria in Solid Tumors (version 1.1). The study used data from the Chang Gung Research Database and the study protocol was approved by the Ethics Committee of Chang Gung Memorial Hospital.

### 2.2. Statistical Analysis

A Mann–Whitney test was used to determine the statistical significance of continuous variables between the two groups and Fisher exact test was used for evaluating the categorical variables. The Kaplan–Meier survival curve was analyzed using the R package *survival*, and the hazard ratio (HR) was analyzed using the Cox regression model. The propensity score-matched analysis was used to balance the clinical characteristics between the treatment groups. Briefly, the afatinib plus bevacizumab and afatinib alone groups served as the dependent variables and the covariates used included age, ECOG PS, stage, *EGFR* mutation subtypes, brain metastasis, liver metastasis, and type of EGFR-TKI administered. The pairs of afatinib plus bevacizumab and afatinib alone individuals with equivalent propensity scores were selected in a 1:3 manner using the R package *MatchIt*. All the reported *p* values were two sided, and a *p* < 0.05 was considered statistically significant. Data were also analyzed using SPSS (version 10.1; SPSS, Chicago, IL, USA).

## 3. Results

### 3.1. Baseline Patient Characteristics

A total of 405 patients were included, of which 367 (90.6%) patients received afatinib treatment alone and 38 (9.4%) patients received afatinib plus bevacizumab. Compared to afatinib alone group, patients who received a combination of bevacizumab were significantly younger (54.6 ± 10.9 vs. 63.9 ± 11.5; *p* < 0.001, Table 1). Patients of the combination group also exhibited a trend of lower rate of stage III disease (0 vs. 6.3%, *p* = 0.150, Table 1). The other clinical features including sex, smoking status, histology, *EGFR* mutation subtypes, and presence of brain and liver metastasis were similar between the two groups.

### 3.2. Efficacies of Bevacizumab in a Propensity Score-Matched Cohort

Propensity score matching was performed in a 1:3 fashion between the afatinib plus bevacizumab and the afatinib alone groups, where a propensity score-matched cohort of 152 patients was achieved in which the afatinib plus bevacizumab group consisted of 34 patients and the afatinib alone group involved 118 patients with balanced clinical characteristics (Table 2). The median follow-up duration was 23.5 months and 37.8 months in the afatinib plus bevacizumab and afatinib alone groups, respectively. Upon data analysis, 19 (55.9%) events of disease progression or death were noted in the afatinib plus bevacizumab group and 93 (78.8%) events were observed in the afatinib alone group. A non-significantly higher objective response was noted in patients who received afatinib plus bevacizumab treatment (82.4% vs. 67.8%; *p* = 0.133, Table 3). The metastatic brain lesions were evaluable for therapeutic response in 13 patients of afatinib plus bevacizumab group and in 33 patients of afatinib monotherapy group, respectively. The intracranial response and intracranial disease control rates were 38.5% and 100.0%, respectively, in the combination group and were 60.6% and 93.9%, respectively, in the single-agent afatinib group. The median PFS (16.1 vs. 15.0 months; log-rank test *p* = 0.500), risk reduction toward disease progression (HR 0.85 [95% CI, 0.52–1.40]; *p* = 0.528), and the 24-month PFS rate (40.9% [95% CI, 25.6% to 65.3%] vs. 32.2% [95% CI, 24.6% to 42.3%], Figure 1A) were similar between the combination and monotherapy groups. The median OS (32.1 vs. 42.0 months; log-rank test *p* = 0.700), risk reduction of death (HR 0.85 [95% CI, 0.42–1.74]; *p* = 0.660), and the 24-month OS rate (73.7% [95% CI, 57.3% to 94.7%] vs. 67.2% [95% CI, 59.1% to 76.5%], Figure 1B) also showed similar results between afatinib plus bevacizumab and afatinib treatment alone group. The post-progression treatments between the two groups were similar (Table 4).

### 3.3. Subgroup Analysis of the Progression-Free and Overall Survival

As PFS did not statistically differ between the afatinib plus bevacizumab and the afatinib alone groups, subgroup analyses of PFS were further explored. An add-on of bevacizumab in male patients (HR 0.56 [95% CI, 0.23–1.35]; *p* = 0.198), patients ≥ 65 years old (HR 0.51 [95% CI, 0.17–1.51]; *p* = 0.222), and patients who had liver metastasis (HR 0.57 [95% CI, 0.19–1.72]; *p* = 0.318, Figure 2) did not exhibit significant PFS improvement. Subgroup analyses of OS also demonstrated no additional benefit of bevacizumab in male patients (HR 0.58 [95% CI, 0.13–2.63]; *p* = 0.486), patients who had *EGFR* L858R mutation (HR 0.65 [95% CI, 0.19–2.17]; *p* = 0.485), and patients who showed absence of brain metastasis (HR 0.53 [95% CI, 0.16–1.75]; *p* = 0.296, Figure 3).

### 3.4. Development of Secondary EGFR T790M Mutation

Of the original cohort of 405 patients, 279 (68.9%) patients underwent disease progression at the time of data analysis. A total of 170 (42.0%) patients received tissue or liquid biopsies for the diagnosis of *EGFR* T790M mutation, of which 85 (50.0%) patients were diagnosed positive for T790M mutation. The T790M positive rate was similar between the afatinib plus bevacizumab and the afatinib alone groups (56.3% vs. 49.4%, Fisher’s exact *p* = 0.794). Clinical factors associated with T790M positivity were assessed by logistic regression. The univariate analysis demonstrated that male sex (OR 0.61 [95% CI, 0.33–1.13]; *p* = 0.122), *EGFR* L858R mutation (OR 0.62 [95% CI, 0.33–1.16]; *p* = 0.137), and *EGFR* uncommon mutation (OR 0.21 [95% CI, 0.03–0.94]; *p* = 0.063) were associated with a lower T790M-positive rate, whereas a PFS longer than 12 months (OR 2.40 [95% CI, 1.27–4.60]; *p* = 0.008) was associated with higher T790M positivity. In multivariate analysis, *EGFR* L858R mutation (OR 0.51 [95% CI, 0.26–0.97]; *p* = 0.044), *EGFR* uncommon mutation (OR 0.14 [95% CI, 0.02–0.64]; *p* = 0.021), and PFS longer than 12 months (OR 2.71 [95% CI, 1.39–5.41]; *p* = 0.004, Table 5) remained independent predictors of secondary T790M positivity.

## 4. Discussion

The present study provided clinical practice-based evidence of first-line afatinib plus bevacizumab treatment from a real-world cohort of Asian NSCLC patients with sensitizing-*EGFR* mutation. The efficacy of this combination demonstrated a trend of higher tumor response, whereas the PFS and OS were similar compared to the single-agent afatinib treatment. In the afatinib plus bevacizumab group, a similar secondary T790M rate was observed compared to the afatinib monotherapy group. In addition, the secondary T790M rate was significantly lower in *EGFR* L858R and uncommon mutation patients but significantly higher in those who experienced a PFS ≥ 12 months of the first-line treatment.

On the basis of real-world practice, we observed that the cohort of patient undergoing treatment of afatinib plus bevacizumab was different from that receiving afatinib monotherapy. Patients of the former cohort were much younger and presented a slightly higher frequency of brain metastasis; physician’s prescription of afatinib plus bevacizumab in real-world practice was largely clinical feature-driven. All these confounding factors have rendered a challenging situation to determine the therapeutic efficacy of bevacizumab add-on to afatinib. Nevertheless, with an appropriate propensity score matching, a balanced characteristic between afatinib plus bevacizumab and single-agent afatinib groups remained achievable to enable a direct outcome analysis.

Previous clinical trials involving Japanese *EGFR*-mutated NSCLC patients demonstrated an add-on of bevacizumab to erlotinib provided a prolonged PFS but not an OS benefit [10,11]. Recently, a similar PFS benefit in patients of *EGFR* mutation receiving bevacizumab and first-generation EGFR-TKI combination was reported in a real-world study by Tsai et al. [26]. In this study, OS benefit was also observed in the *EGFR* L858R mutation patients receiving bevacizumab and EGFR-TKI combination. In a recent study from an Italian NSCLC cohort involving both common and uncommon sensitizing *EGFR* mutations, a significant PFS benefit and a trend of improved OS toward bevacizumab plus erlotinib compared to erlotinib monotherapy groups were also noted [27]. In these erlotinib-based trials [10,27], smoking history did not seem to be a factor that impacted the magnitude of benefit of bevacizumab treatment. In the present analysis, an afatinib-based study, a lack of interaction of bevacizumab treatment with smoking status was similarly observed.

In contrary to patients receiving erlotinib-based treatment, our results suggested that patients who underwent afatinib-based treatment received no additional benefit from a bevacizumab add-on. This finding may be related to a higher tumor response by single-agent afatinib compared to the single-agent first-generation EGFR-TKI (73% vs. 56%) observed previously [1], where a deeper or more durable response may reduce the synergistic effect of anti-angiogenesis agents. On the other hand, the mechanism of action of the two therapies may have a certain overlap. Previous studies have revealed that neuregulin-dependent ErbB3 and ErbB4 signaling in cancer cells contributed to VEGF-mediated angiogenesis and anti-apoptosis [28,29,30,31]. Furthermore, non-cancerous stromal cell- or vascular endothelial cell-derived neuregulin also promoted angiogenesis and VEGF expression via ErbB3 and ErbB4 in an autocrine or paracrine manner [32,33]. Given that afatinib suppresses the activation of pan-ErbB family with high potency [34], the VEGF-driven pathobiology may be partly alleviated in tumor microenvironment and, thus, reduced the additional efficacy of VEGF-targeting agents.

In a previous animal model, treatment of bevacizumab has demonstrated activity to mice xenograft bearing *EGFR* T790M mutation [21]. In the BELIEF trial, patients of de novo T790M mutation identified by a sensitive peptide nucleic acid-clamping PCR assay also exhibited a prolonged PFS to erlotinib and bevacizumab combination [23]. These findings led to the hypothesis that an anti-angiogenesis agent and EGFR-TKI combination may contain the emergence of drug-resistant secondary T790M mutation. In the NEJ 026 study, the secondary T790M identified in erlotinib plus bevacizumab and single-agent erlotinib groups were 20.8% and 19.0%, respectively [24]. Another trail involving ramicirumab and erlotinib combination, the RELAY study, also showed similar T790M rates of 43% in the combination arm and of 47% in the single-agent erlotinib arm [25]. Interestingly, in a previous transgenic mouse model, a combination of afatinib and bevacizumab effectively suppressed tumor bearing *EGFR* 19 deletion/T790M and L858R/T790M mutations compared to treatment by either drug alone [35]. Nevertheless, in the present analysis, secondary T790M rates between the afatinib plus bevacizumab and the single-agent afatinib groups remain similar. Overall, whether the combination of afatinib and bevacizumab alters the development of secondary T790M requires further investigation.

The inherent limitation of the present study, firstly, is its retrospective nature. Secondly, heterogeneous clinical features were observed between the original afatinib plus bevacizumab and afatinib monotherapy groups. However, with the proper adjustment of propensity score matching, this heterogeneity is maximally moderated to enable a direct comparison. Thirdly, the recent administration of third-generation osimertinib monotherapy as well as the combination of osimertinib and bavaciumab also challenges the role of afatinib on the treatment of advanced *EGFR*-mutant NSCLC [36,37]. However, a recent Japanese real-world cohort of NSCLC patients demonstrated an improved OS of front-line afatinib compared to osimertinib treatment [38] and the true benefit of adding bevacizumab to front-line osimertinib remains largely unsettled based on some small early-phase trials [37].

In conclusion, this work presented a real-world cohort of NSCLC patients with *EGFR* mutation properly adjusted for clinical biases to compare the efficacy of afatinib plus bevacizumab and single-agent afatinib treatment. The result demonstrated that patients receiving bevacizumab combination had a similar survival outcome and secondary T790M incidence as those receiving afatinib monotherapy. Further investigation of clinical sub-cohort that benefits from bevacizumab treatment is warranted in afatinib-treated *EGFR*-mutant patients.

## Figures and Tables

**Figure 1 cancers-14-00316-f001:**
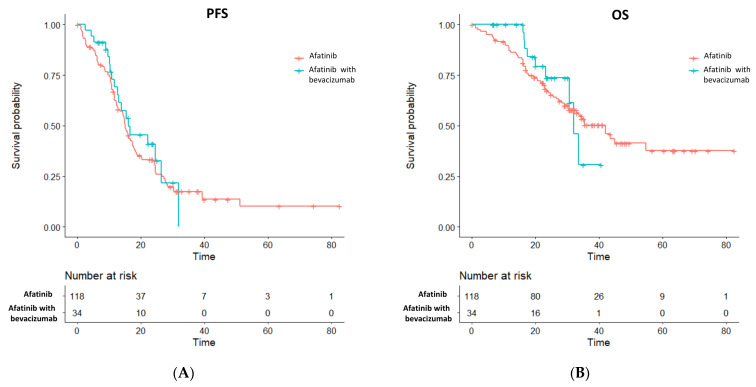
Survival outcome of (**A**) PFS and (**B**) OS between the afatinib plus bevacizumab and the single-agent afatinib groups in the propensity score-matched cohort.

**Figure 2 cancers-14-00316-f002:**
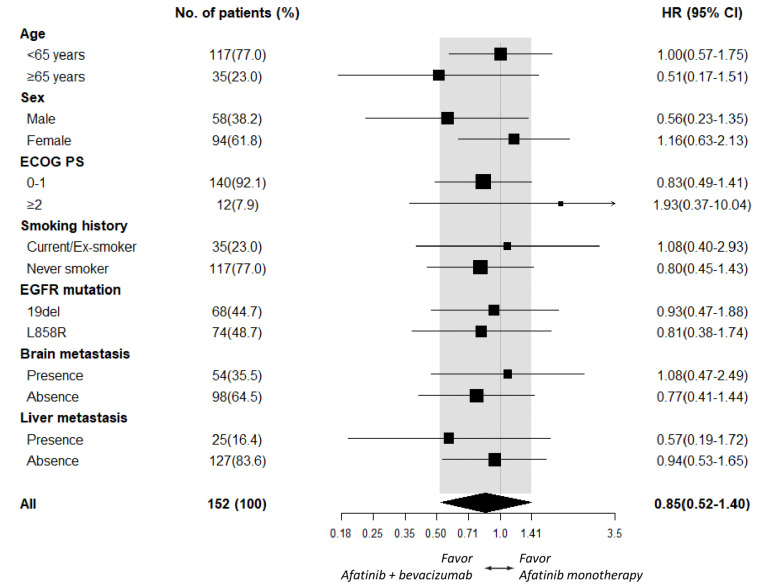
Subgroup analysis of PFS between the afatinib plus bevacizumab and the afatinib monotherapy groups.

**Figure 3 cancers-14-00316-f003:**
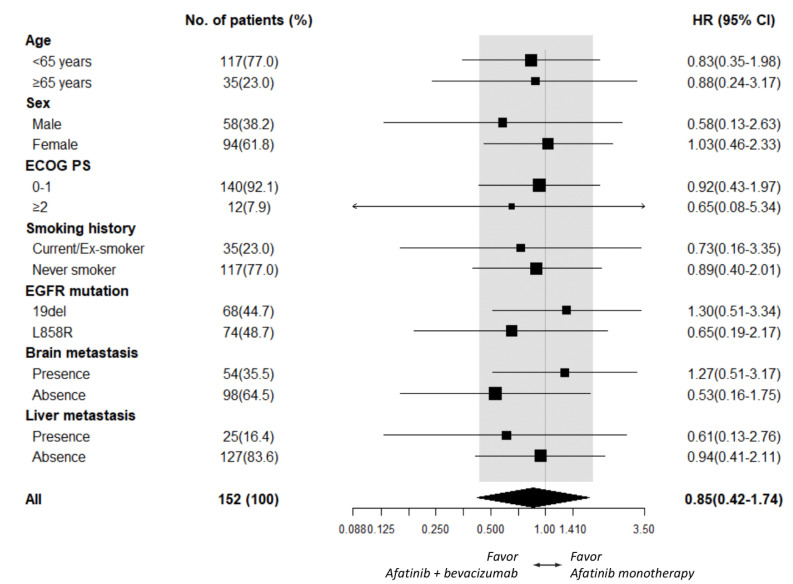
Subgroup analysis of OS between the afatinib plus bevacizumab and the afatinib monotherapy groups.

**Table 1 cancers-14-00316-t001:** Overall patient characteristics.

Varialbles	Total (%) N = 405	Afatinib Plus Bevacizumab (%) N = 38	Afatinib Alone (%) N = 367	*p* Value
Age (mean ± SD)		54.6 ± 10.9	63.9 ± 11.5	<0.001
Age ≥ 65	180 (44.4)	9 (23.7)	171 (46.6)	0.009
ECOGPS 0-1	370 (91.4)	36 (94.7)	334 (91.0)	0.760
Gender				
Male	161 (39.8)	15 (39.5)	146 (39.8)	1.000
Current/ex-smoker	83 (20.5)	8 (21.1)	75 (20.4)	1.000
Histology				
Adenocarcinoma	399 (98.5)	37 (97.4)	362 (98.6)	0.449
Others	6 (1.5)	1 (2.6)	5 (1.4)	
*EGFR* mutation				
L858R	185 (45.7)	19 (50.0)	166 (45.2)	0.887
19deletion	196 (48.4)	17 (44.7)	179 (48.8)	
Uncommon	24 (5.9)	2 (5.3)	22 (6.0)	
Disease Stage				
III	23 (5.7)	0	23 (6.3)	0.150
IV	382 (94.3)	38 (100.0)	344 (93.7)	
Site of Metastasis				
Brain	121 (29.9)	14 (36.8)	107 (29.2)	0.354
Liver	50 (12.3)	6 (15.8)	44 (12.0)	0.445

**Table 2 cancers-14-00316-t002:** Propensity score-matched cohort.

Varialbles	Total (%) N = 152	Afatinib and Bevacizumab Combination (%) N = 34	Afatinib Alone (%) N = 118	*p* Value
Age (mean ± SD)		56.0 ± 10.8	58.0 ± 9.2	0.341
Age ≥ 65	35 (23.0)	9 (26.5)	26 (22.0)	0.645
ECOG PS 0-1	140 (92.1)	32 (94.1)	108 (91.5)	1.000
Gender				
Male	58 (38.2)	14 (41.2)	44 (37.3)	0.693
Current/ex-smoker	35 (23.0)	7 (20.6)	28 (23.7)	0.819
Histology				
Adenocarcinoma	150 (98.7)	33 (97.1)	117 (99.2)	0.399
Others	2 (1.3)	1 (2.9)	1 (0.8)	
*EGFR* mutation				
L858R	74 (48.7)	16 (47.1)	58 (49.2)	0.952
19deletion	68 (44.7)	16 (47.1)	52 (44.1)	
Uncommon	10 (6.6)	2 (5.8)	8 (6.7)	
Disease Stage				
IV	152 (100.0)	34 (100.0)	118 (100.0)	1.000
Site of Metastasis				
Brain	54 (35.5)	13 (38.2)	41 (34.7)	0.839
Liver	25 (16.4)	6 (17.6)	19 (16.1)	0.798
Afatinib dose reduction	51 (33.6)	12 (35.3)	39 (33.1)	0.838

**Table 3 cancers-14-00316-t003:** Objective response in the propensity score-matched cohort.

Variables, n (%)	Afatinib Plus Bevacizumab N = 34	Afatinib Alone N = 118
Response		
No. of patients	28	80
% (95% CI)	82.4 (65.4–93.2)	67.8 (58.6–76.1)
Complete response—No. (%)	0	2 (1.7)
Partial response—No. (%)	28 (82.4)	78 (66.1)
Stable disease—No. (%)	4 (11.8)	25 (21.2)
Progression disease—No. (%)	2 (5.8)	13 (11.0)
Median duration of response—month (95% CI)	22.2 (13.0–not reach)	17.8 (15.7–24.3)

**Table 4 cancers-14-00316-t004:** Post-progression treatment.

Treatments, n (%)	Afatinib Plus Bevacizumab N = 34	Afatinib Alone N = 118
Third-generation EGFR-TKI	12 (35.3)	33 (28.0)
Chemotherapy	9 (26.5)	38 (32.2)
Immune checkpoint inhibitor	2 (5.9)	6 (5.1)
Other TKIs	3 (8.8)	16 (13.6)

**Table 5 cancers-14-00316-t005:** Factors associated with T790M positivity.

Variables	Univariate Analysis		Mutivariate Analysis	
	Odd Ratio (95% C.I.)	*p*-Value	Odd Ratio (95% C.I.)	*p*-Value
Age ≥ 65	0.95 (0.51–1.77)	0.874	--	--
Male	0.62 (0.33–1.13)	0.122	0.59 (0.31–1.13)	0.111
ECOG PS 0-1	1.00 (0.27–3.72)	1.000	--	--
Current/ex-smoker	0.60 (0.29–1.21)	0.157	--	--
*EGFR* L858R mutation	0.62 (0.33–1.16)	0.137	0.51 (0.26–0.97)	0.044
*EGFR* uncommon mutation	0.21 (0.03–0.94)	0.063	0.14 (0.02–0.64)	0.021
Brain metastasis	1.02 (0.52–1.96)	0.959	--	--
Liver metastasis	0.90 (0.35–2.25)	0.816	--	--
PFS ≥ 12 months	2.40 (1.27–4.60)	0.008	2.71 (1.39–5.41)	0.004

## Data Availability

The datasets generated and/or analysed in the current study are available from the corresponding author on reasonable request.

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
