# Peer review of "Afatinib Treatment Alone or with Bevacizumab in a Real-World Cohort of Non-Small Cell Lung Cancer Patients with Epidermal Growth Factor Receptor Mutation"

_cancers, 2022, doi:10.3390/cancers14020316_

Round 1

Reviewer 1 Report

The Authors reported the results of a large retrospective study evaluating the role of first-line afatinib alone or with bevacizumab in advanced NSCLC patients with EGFR mutation. They found that bevacizumab treatment  showed moderate  efficacy  in real-world, afatinib-treated  NSCLC patients  with EGFR-sensitizing mutation. The results are interesting and well presented. I suggest to comment in the discussion the lack of interaction with smoker status, reported in other studies with bevacizumab and erlotinib, including the most recent trial presented by Piccirillo et al (cited as ref. 28).

Author Response

Reviewer 1:

The Authors reported the results of a large retrospective study evaluating the role of first-line afatinib alone or with bevacizumab in advanced NSCLC patients with EGFR mutation. They found that bevacizumab treatment showed moderate efficacy in real-world, afatinib-treated NSCLC patients with EGFR-sensitizing mutation. The results are interesting and well presented. I suggest to comment in the discussion the lack of interaction with smoker status, reported in other studies with bevacizumab and erlotinib, including the most recent trial presented by Piccirillo et al (cited as ref. 28).

Reply to comment:

We thank to the reviewer’s comment and have revised this part as in page 8 line 7 to 11 as following:

“In these erlotinib-based trials [10, 28], smoking history did not seem to be a factor that impacted the magnitude of benefit of bevacizumab treatment. In present analysis, an afatinib-based study, the lack of interaction of bevacizumab treatment with smoking status was similarly observed.”

Reviewer 2 Report

Authors retrospectively analyzed 454 advanced EGFR mutated NSCLC patients (152 patients using propensity-score matched analysis) in terms of PFS, OS, and emergence of secondary T790M mutation. They concluded that combination of afatinib with bevacizumab showed moderate efficacy in real-world in NSCLC patients with EGFR-sensitizing mutation. The T790M positive rate was similar between the afatinib plus bevacizumab and the afatinib alone groups. In multivariate analysis, EGFR L858R mutation (OR 0.51), EGFR uncommon mutation (OR 0.14) and PFS longer than 12 months (OR 2.71) remained independent predictors of secondary T790M positivity.

The paper was well organized and discussed, however, there were only a few new findings. Thus, I will recommend that this manuscript might be submitted as a brief report.

Line 92: Afatinib -> afatinib

Author Response

Reviewer 2:

Authors retrospectively analyzed 454 advanced EGFR mutated NSCLC patients (152 patients using propensity-score matched analysis) in terms of PFS, OS, and emergence of secondary T790M mutation. They concluded that combination of afatinib with bevacizumab showed moderate efficacy in real-world in NSCLC patients with EGFR-sensitizing mutation. The T790M positive rate was similar between the afatinib plus bevacizumab and the afatinib alone groups. In multivariate analysis, EGFR L858R mutation (OR 0.51), EGFR uncommon mutation (OR 0.14) and PFS longer than 12 months (OR 2.71) remained independent predictors of secondary T790M positivity.

The paper was well organized and discussed, however, there were only a few new findings. Thus, I will recommend that this manuscript might be submitted as a brief report.

Reply to comment:

We thank to reviewer’s comments. Currently, there is very limited data about the comparison between treatments using 2nd generation EGFR TKI alone or with anti-angiogenesis in advanced NSCLC patients of EGFR mutation and thus we believe this would be the merit of the present study to the journal’s audience.

Line 92: Afatinib -> afatinib

Reply to comment:

We have corrected the typo at page 2 line 46.

Reviewer 3 Report

In this study, the authors compared treatment efficacies of afatinib alone vs. afatinib plus bevacizumab as 1st line treatment in NSCLC patients with sensitive EGFR mutation. The authors used propensity score matching technique to balance the clinical characteristics between the treatment groups. In general, the reviewer thinks the study is well performed. However, the reviewer raises some important comments as summarized below.

  1. Dose reduction is often needed during treatment with afatinib. Because dose reduction may be potentially related to the efficacy, the reviewer suggests to compare "dose reduction rates" between afatinib group vs. afatinib plus bevacizumab group in the propensity score matched cohort.
  2. In the subgroup analyses, the authors described that "subgroup analyses of PFS suggested an improved outcome with an add-on of bevacizumab in male patients (HR 0.56 [95% CI, 0.23-1.35]; p=0.198), patients ≥ 65 year old (HR 0.51 [95% CI, 0.17-1.51]; p=0.222), and patients who had liver metastasis (HR 0.57 [95% CI, 0.19-1.72]; p=0.318, Figure 2)." There are similar descriptions for OS analyses. However, as shown in Figures 2 and 3, the reviewer thinks it is inappropriate to say "improved outcome with an add-on of bevacizumab", because the difference was not statistically significant at all. The reviewer thinks the conclusion of these subgroup analyses should be "no significant differences between treatment groups even in the subgroup analyses".
  3. The reviewer also suggests to compare CNS response rates between treatment groups in patients with brain metastases.

Author Response

Reviewer 3:

In this study, the authors compared treatment efficacies of afatinib alone vs. afatinib plus bevacizumab as 1st line treatment in NSCLC patients with sensitive EGFR mutation. The authors used propensity score matching technique to balance the clinical characteristics between the treatment groups. In general, the reviewer thinks the study is well performed. However, the reviewer raises some important comments as summarized below.

  1. Dose reduction is often needed during treatment with afatinib. Because dose reduction may be potentially related to the efficacy, the reviewer suggests to compare "dose reduction rates" between afatinib group vs. afatinib plus bevacizumab group in the propensity score matched cohort.

Reply to comment:

We agree with the reviewer that dose reduction rate should be taken into account in the propensity score matched cohort. Overall, 51 (33.6%) patients of the propensity score matched cohort underwent dose reduction; with 12 (35.3%) patients in the combination group and 39 (33.1%, fisher’s exact p=0.838) patients in the monotherapy group, respectively. We have provided this data to clarify the influence of this parameter in the revised Table 2.

  1. In the subgroup analyses, the authors described that "subgroup analyses of PFS suggested an improved outcome with an add-on of bevacizumab in male patients (HR 0.56 [95% CI, 0.23-1.35]; p=0.198), patients ≥ 65 year old (HR 0.51 [95% CI, 0.17-1.51]; p=0.222), and patients who had liver metastasis (HR 0.57 [95% CI, 0.19-1.72]; p=0.318, Figure 2)." There are similar descriptions for OS analyses. However, as shown in Figures 2 and 3, the reviewer thinks it is inappropriate to say "improved outcome with an add-on of bevacizumab", because the difference was not statistically significant at all. The reviewer thinks the conclusion of these subgroup analyses should be "no significant differences between treatment groups even in the subgroup analyses".

   Reply to comment:

We agree with the reviewer’s comment and have revised the Result section in page 5 line 17 to 26 as following:

“As PFS did not statistically differ between the afatinib plus bevacizumab and the afatinib alone groups, subgroup analyses of PFS was further explored. An add-on of bevacizumab in male patients (HR 0.56 [95% CI, 0.23-1.35]; p=0.198), patients ≥ 65 year old (HR 0.51 [95% CI, 0.17-1.51]; p=0.222), and patients who had liver metastasis (HR 0.57 [95% CI, 0.19-1.72]; p=0.318, Figure 2) did not exhibit significant PFS improvement. Subgroup analyses of OS also demonstrated no additional benefit of bevacizumab in male patients (HR 0.58 [95% CI, 0.13-2.63]; p=0.486), patients who had EGFR L858R mutation (HR 0.65 [95% CI, 0.19-2.17]; p=0.485) and patients who were absence of brain metastasis (HR 0.53 [95% CI, 0.16-1.75]; p=0.296, Figure 3).”

  1. The reviewer also suggests to compare CNS response rates between treatment groups in patients with brain metastases.

Reply to comment:

We have supplied the data of intracranial response in the Result section page 4 line 13 to 17 as following:

“The metastatic brain lesions were evaluable for therapeutic response in 13 patients of afatinib plus bevacizumab group and in 33 patients of afatinib monotherapy group, respectively. The intracranial response and intracranial disease control rates were 38.5% and 100.0% respectively in the combination group and were 60.6% and 93.9% respectively in the single-agent afatinib group.”

Round 2

Reviewer 2 Report

I understand that  there is very limited data about the comparison between treatments using 2nd generation EGFR TKI alone or with anti-angiogenesis in advanced NSCLC patients of EGFR mutation. However, this manuscript did not have new findings in EGFR-TKI plus anti-angiogenesis.